# Serum and plasma brain-derived neurotrophic factor concentration are elevated by systemic but not local passive heating

Takahiro Ogawa[1◉], Sven P. Hoekstra[2◉], Yoshi-Ichiro Kamijo[1‡], Victoria L. Goosey-Tolfrey[2‡], Jeremy J. Walsh[3‡], Fumihiro Tajima F.[1,2‡], Christof A. Leicht[2]*

1 Department of Rehabilitation Medicine, Wakayama Medical University, Wakayama, Japan, 2 The Peter Harrison Centre for Disability Sport, Loughborough University, Loughborough, United Kingdom, 3 Department of Kinesiology, McMaster University, Hamilton, Ontario, Canada

◉ These authors contributed equally to this work.
‡ These authors also contributed equally to this work.
* c.a.leicht@lboro.ac.uk

**Data Availability Statement:** The data set is publicly available at https://figshare.com/s/d982de1e275b91635ceb.

## Abstract

Brain-derived neurotrophic factor (BDNF) plays a key role in neuronal adaptations. While previous studies suggest that whole-body heating can elevate circulating BDNF concentration, this is not known for local heating protocols. This study investigated the acute effects of whole-body versus local passive heating on serum and plasma BDNF concentration. Using a water-perfused suit, ten recreationally active males underwent three 90 min experimental protocols: heating of the legs with upper-body cooling (LBH), whole-body heating (WBH) and a control condition (CON). Blood samples were collected before, immediately after and 1 h post-heating for the determination of serum and plasma BDNF concentration, platelet count as well as the BDNF release per platelet. Rectal temperature, cardiac output and femoral artery shear rate were assessed at regular intervals. Serum and plasma BDNF concentration were elevated after WBH (serum: 19.1±5.0 to 25.9±11.3 ng/ml, plasma: 2.74±0.9 to 4.58±2.0; $p<0.044$), but not LBH (serum: 19.1±4.7 to 22.3±4.8 ng/ml, plasma: 3.25±1.13 to 3.39±0.90 ng/ml; $p>0.126$), when compared with CON (serum: 18.6±6.4 to 16.8±3.4 ng/ml, plasma: 2.49±0.69 to 2.82±0.89 ng/ml); accompanied by an increase in platelet count ($p<0.001$). However, there was no change in BDNF content per platelet after either condition ($p = 0.392$). All physiological measures were elevated to a larger extent after WBH compared with LBH ($p<0.001$), while shear rate and rectal temperature were higher during LBH than CON ($p<0.038$). In conclusion, WBH but not LBH acutely elevates circulating BDNF concentration. While these findings further support the use of passive heating to elevate BDNF concentration, a larger increase in shear rate, sympathetic activity and/or rectal temperature than found after LBH appears needed to induce an acute BDNF response by passive heating.

**Funding:** This project was supported by the Kyoten Research Center of Sports for Persons with Impairments. The funders were not involved in any stage of the research process.

**Competing interests:** The authors have declared that no competing interests exist.

## Introduction

Brain-derived neurotrophic factor (BDNF) is a member of the neurotrophin family that plays a key role in neuroplasticity, learning and memory as well as metabolic regulation [1]. BDNF knock-out mice show impaired spatial learning, reduced survival of neurons in the hippocampus [2] and develop obesity [3]. Furthermore, blocking the BDNF receptor tropomyosin receptor kinase B (TrkB) attenuates improvements in cognitive function following exercise training in rats [4]. The strong correlation of BDNF expressed in brain structures such as the hippocampus with circulating BDNF concentration provides the opportunity to investigate BDNF expression in humans [5]. As such, an increasing number of studies have investigated strategies to elevate circulating BDNF concentration as well as their potential to treat or prevent conditions related to neuronal impairments such as Alzheimer's disease, depression and schizophrenia [6].

Acute and chronic aerobic exercise can elevate circulating BDNF concentration in humans [7]. However, physical exercise is not universally accessible to all members of society, due to for instance disability, chronic disease or cognitive impairments. Passive heat treatment is a promising intervention strategy that may benefit systemic health, especially in situations where exercise is not accessible [8, 9]. In the context of elevating BDNF concentration, a single bout of aerobic exercise in the heat induces a larger acute BDNF response compared to exercise in a thermoneutral environment [10]. Moreover, 20 min of whole-body immersion in 42˚C water also acutely elevates serum BDNF concentration [11]. In support of this acute effect, a recent trial found that 10 weeks of repeated head-out, dry hyperthermic exposure significantly increased serum BDNF concentration in young adults [12]. However, as with exercise, the cardiovascular and heat strain induced by whole-body passive heating may preclude some individuals from engaging in this activity [13]. For instance, people with chronic heart failure may be advised against engagement in whole-body heating due to the attendant cardiovascular strain [14], while the impaired thermoregulation in older adults and persons with type II diabetes mellitus may place them at an increased risk for heat-illness during intense heat stress [15, 16]. As such, prior to promoting passive heating as a strategy to stimulate BDNF-mediated improvements in cognitive function, more accessible and physiologically less strenuous protocols may provide an additional tool to promote health in persons for whom whole-body heating may be contraindicated.

The primary cellular sources of circulating BDNF are suggested to be the brain, vascular endothelial cells, and peripheral blood mononuclear cells. A small proportion of circulating BDNF is unbound and freely interacts with TrkB, whereas the majority of circulating BDNF is bound to platelets [17]. Increased BDNF in response to hyperthermia may be mediated through multiple mechanisms, including increased in shear stress, stimulating BDNF release by endothelial cells and platelets [18, 19]; an increase in the permeability of the blood-brain-barrier with an increase in body temperature, increasing the contribution of BDNF from the brain [11]; or a sympathetic activation-mediated increase in the release of BDNF containing platelets from the spleen (thrombocytosis) [20]. These suggested drivers of the acute BDNF response following heat stress indicate that systemic hyperthermia may not be essential to elevate BDNF concentration, as for instance blood flow can also be increased through localised heating [21]. Apart from a reduced cardiovascular strain, local passive heating has recently also been shown to result in more favourable perceptual responses compared with whole-body passive heating [22]; potentially positively affecting uptake and adherence to passive heating interventions [23]. However, the effect of local heating on the circulating BDNF concentration has yet to be determined.

Therefore, this study compared the effects of acute whole-body heating (WBH) versus lower-limb heating in combination with cooling of the upper body (LBH) on circulating BDNF and cardiovascular strain in young, healthy adults. It was hypothesised that the LBH would evoke lower cardiovascular strain compared with WBH, but would nonetheless elevate circulating BDNF concentration to a similar magnitude as WBH due to increases in shear stress and sympathetic nerve activity.

## Materials and methods

Ten healthy young males (age: 24±3 yrs; height: 184±6 cm; body mass: 80±15 kg; BMI: 23.0 ±4.9 kg/m$^2$; body fat percentage: 15.7±4.2%) participated in this study after providing written informed consent. Exclusion criteria were smoking and the use of anti-inflammatory medication. This study reports secondary findings from a larger trial that investigated the effect of LBH versus WBH on inflammation, glycaemic and perceptual responses [21]. As such, the participants and heating protocols described herein are identical to Hoekstra et al. [21]. The study procedures were approved by the ethics committee of Loughborough University (project code: R19-P084), according to the declaration of Helsinki.

### Study design

Participants visited the laboratory following an overnight fast on three occasions, separated by at least 72 h. Participants avoided exercise and the consumption of caffeine and alcohol on the day prior to their laboratory visits. In addition, they monitored their food and drink consumption prior to the first laboratory visit and adhered to the same diet on the day before the following visits. The heating protocols used have been described in detail previously [21]. Briefly, body temperature was manipulated using a water-perfused suit (Med-Eng, Ottawa, Canada), with separate controls for the lower and upper body segments. Participants undertook three 90 min experimental conditions in a randomised order: 1) whole-body heating (WBH), where 50˚C water was perfused through the upper and lower body part of the suit; 2) lower-body heating with simultaneous cooling of the upper body (LBH), where 50˚C water was perfused through the lower body part of the suit and upper-body cooling was applied by cool packs and 2˚C water perfused through the upper body part of the suit; 3) a control condition (CON), where 36˚C water was perfused through both parts of the suit. Ambient temperature and relative humidity in the laboratory were 24.4±0.6˚C and 44±8% during CON, 24.6±0.7˚C and 43 ±9% during LBH, and 24.5±1.0˚C and 46±8% during WBH (*p*>0.452).

### Procedures

Height, body mass and skinfold thickness (biceps, triceps, supra-iliac and subscapular) were assessed on the first visit [24]. Thereafter, participants applied a rectal temperature probe 10 cm beyond the anal sphincter for the measurement of rectal temperature ($T_{rec}$). A zero-heat flux temperature sensor (Bair Hugger, 3M, Minnesota, USA) was placed on the skin at the muscle belly of the vastus lateralis to measure deep tissue temperature ($T_{dt}$) [25]. The sensor was covered by a small Tupperware box to limit any thermal effects of the water-perfused suit. Tympanic temperature was measured by a temperature sensor (Squirrel, Grant Instruments, Shepreth, UK) worn throughout the session, secured in the left ear by cotton wool and industrial headphones. Nude body mass was then assessed to the nearest 0.1 kg (Seca 284, Hamburg, Germany). Skin thermometers were fitted on the chest, triceps, thigh and calf (Squirrel, Grant Instruments, Shepreth, UK), and a cannula was inserted into an antecubital vein. Participants rested in a supine position for 60 min wearing shorts and a T-shirt. At the end of the rest period, temperature measures, heart rate (HR; Polar, Kempele, Finland) and arterial blood

pressure were recorded. Blood pressure was measured in duplicate using an automated cuff (Microlife, Cambridge, UK) at the brachial artery in the left arm. Thereafter, participants put on the water-perfused suit for the experimental condition. Physiological measures were assessed every 15 min, and at 30 min and 60 min post-session. After removing the water-perfused suit, a blood sample was collected, and the participant remained supine for an additional 30 min. Thereafter, nude body mass was measured. The final blood sample was collected 60 min post-session.

Participants were provided with water during the sessions to offset weight loss through sweating. During WBH, 150 ml of water was provided prior to and at 15 min intervals during heating. For LBH, 100 ml water was provided before and at the end of the heating protocol, while during CON 50 ml was provided directly following the 90 min session. Heart rate and systolic blood pressure were used to calculate rate pressure product as a measure of cardiac strain [26]; heart rate and $T_{rec}$ were used to calculate the physiological strain index [27].

## Ultrasonography

Brachial and common femoral artery blood flow as well as cardiac output were assessed pre and directly post-heating as described in Hoekstra et al. [21]. Briefly, arterial blood flow and shear rate were assessed by ultrasonography (GE Healthcare, Chicago IL, USA) in duplicate at each time point and in the Doppler mode, which records arterial images and blood velocity signals simultaneously. Non-blinded measurements and analyses were performed by the same experienced ultrasonographer (T.O., 20 years of experience in ultrasonography), with a CV of 3.7% for femoral artery blood flow and 5.4% for brachial artery blood flow based on the baseline data obtained in CON. A16 MHz linear array transducer was used, and images were acquired at an insonation angle of 60˚ for 10 heart cycles. Arterial diameter was measured by identifying the adventitial border of the near and far walls of the artery using the built-in caliper function on the ultrasound unit. Two caliper measurements per image were taken and averaged to yield a diameter value. Blood flow was calculated as the product of the mean blood velocity during a cardiac cycle and the cross-sectional area of the vessel. Shear rate was calculated by the following formula: [4*(mean blood velocity/vessel diameter)] [28]. Vascular conductance in the femoral artery was determined as femoral artery blood flow/mean arterial pressure; the latter calculated as [(systolic blood pressure + (diastolic blood pressure*2))/3] [29].

Stroke volume and cardiac output were also measured by Doppler ultrasound (GE Healthcare, Chicago IL, USA), via the Doppler method [30]. Using a M5S transducer and keeping the participant in the left lateral decubitus position, left ventricular outflow tract diameter was measured using a parasternal long-axis view, while left ventricular flow (velocity time integral) was acquired in the 3- or 5- chamber view obtained immediately proximal to the aortic valve. These two variables were then used to calculate stroke volume. All cardiac measurements and analyses were performed by the same unblinded ultrasonographer (T.O.). Cardiac output was obtained as the product of stroke volume and HR.

## Blood analyses

Blood was drawn into a $K_3$EDTA and serum monovette. Plasma samples were centrifuged immediately for 10 min at 2360 *g* and 4˚C. Serum samples underwent the same centrifugation procedure after they were allowed to clot for 30 min at room temperature. Plasma and serum aliquots were stored at -80˚C until batch analysis. Enzyme-linked immunosorbent assays were used to determine serum and plasma BDNF concentration (R&D systems, Abingdon, UK) as well as plasma adrenaline concentration (Tecan UK Ltd, Reading, UK). Serum samples were

diluted 30-fold. Haemoglobin concentrations and whole blood counts were assessed by a Yumizen H500 (Horiba Medical, Montpellier, France) automated analyser. Haematocrit, determined in duplicate using a microcentrifuge, and haemoglobin were used to correct BDNF concentrations and heamatological parameters for changes in plasma and blood volume, respectively [31]. The BDNF content in platelets was calculated according to the method postulated by Lommatszsch et al. [32]: [(serum BDNF concentration–plasma BDNF concentration)/platelet count].

## Statistical analysis

All data are presented as mean ± SD. Normality and sphericity were checked by the Shapiro Wilk and Mauchley's test, respectively. Changes in physiological, thermoregulatory and BDNF data were analysed by 2-way repeated measures ANOVA, with Fisher's LSD tests used for post-hoc comparisons [33]. Data of WBH were used to calculate Pearson's correlations between the change in serum, plasma and platelet BDNF, and $T_{core}$, femoral artery shear rate and HR were calculated. The 24th version of SPSS (Chicago IL, USA) was used for all analyses and significance was accepted at $p < 0.05$.

## Results

### Thermoregulatory measures

The thermoregulatory responses to the three experimental conditions are shown in Table 1. Rectal temperature at the end of WBH was 38.6±0.4°C, while its rise was reduced by upper-body cooling ($T_{rec}$ end LBH: 37.1±0.3°C; time x condition $p < 0.001$). Nevertheless, $T_{rec}$ at the end of LBH was higher than CON ($T_{rec}$ end CON: 36.7±0.2°C; $p = 0.001$). There was an effect of time ($p < 0.001$) and time x condition interaction ($p < 0.001$) for tympanic temperature, with higher values in WBH compared with the other conditions throughout the session ($p < 0.006$). There was no difference in tympanic temperature between LBH and CON at any time point ($p > 0.123$). Deep tissue temperature was elevated to a larger extent by WBH compared with the other conditions (WBH: from 35.4±0.68 to 38.7±0.48, LBH: from 35.7±0.62 to 37.3±0.42, CON: from 35.3±0.36 to 36.1±0.24; time x condition $p < 0.001$), while $T_{dt}$ during LBH was also higher compared with CON from 30 min onwards ($p < 0.001$). There was no difference in the change in body mass between conditions (CON: 0.08±0.07 kg, LBH: 0.07±0.21 kg, WBH: 0.18 ±0.32 kg; $p = 0.321$).

### Cardiovascular measures

Cardiovascular outcome measures in response to the three experimental conditions are shown in Table 2. Blood flow in the common femoral artery was elevated to a larger extent by WBH compared with the other conditions (time x condition $p < 0.001$), while it was also higher during LBH compared with CON at 45 min and 90 min ($p = 0.002$ at both time points). Shear rate in the femoral artery was higher in WBH compared with LBH ($p < 0.001$), and in LBH compared with CON at 45 min and 90 min ($p < 0.038$). There was an effect of time ($p < 0.001$), condition ($p < 0.001$) and a time x condition interaction ($p < 0.001$) for rate pressure product, such that this was higher throughout WBH compared with the other conditions ($p < 0.001$), while there was no difference between LBH and CON ($p > 0.127$). An effect of time ($p < 0.001$), condition ($p < 0.001$) and a time x condition interaction ($p < 0.001$) was present for the physiological strain index. The physiological strain index was higher throughout WBH compared with both other conditions ($p < 0.001$), while it was also higher in LBH compared with CON at 45 min and 90 min ($p < 0.006$).

**Table 1. Thermoregulatory responses to the three experimental conditions.** Values are expressed as mean ± SD (N = 10).

| Parameter | Condition | | | Time | |
|---|---|---|---|---|---|
| | | Pre | 45 min | End | P+30 min |
| $T_{rec}$ (˚C)^ | CON | 36.6±0.4 | 36.6±0.2 | 36.7±0.2 | 36.7±0.2 |
| | LBH | 36.7±0.2 | 36.9±0.3 | 37.1±0.3* | 36.5±0.3 |
| | WBH | 36.8±0.3 | 37.4±0.3# | 38.6±0.4# | 37.3±0.2 |
| $T_{tympanic}$ (˚C)^ | CON | 35.4±0.4 | 35.7±0.4 | 35.8±0.4 | 35.5±0.3 |
| | LBH | 35.1±0.6 | 35.7±0.6 | 35.8±0.7 | 34.9±0.5 |
| | WBH | 35.3±0.6 | 37.0±0.3# | 38.2±0.4# | 35.2±0.6 |
| $T_{thigh}$ (˚C)^ | CON | 31.6±1.0 | 34.0±0.6 | 34.0±0.6 | 32.6±0.9 |
| | LBH | 31.5±1.0 | 37.8±1.6* | 38.0±1.3* | 33.3±1.3 |
| | WBH | 32.4±1.1 | 38.6±0.7* | 39.3±0.8# | 33.4±1.0 |
| $T_{calf}$ (˚C)^ | CON | 33.2±0.9 | 34.0±1.0 | 34.0±1.0 | 33.2±1.0 |
| | LBH | 33.1±0.9 | 38.2±1.1* | 38.5±1.1* | 35.4±1.0* |
| | WBH | 33.4±0.9 | 38.7±1.9* | 39.7±1.8* | 35.4±1.6* |
| $T_{arm}$ (˚C)^ | CON | 32.1±0.9 | 34.3±0.7 | 34.6±0.6 | 33.4±0.9 |
| | LBH | 32.2±0.8 | 23.1±4.0* | 21.2±3.4* | 26.8±2.2* |
| | WBH | 32.1±0.7 | 38.5±0.8# | 39.3±0.9# | 35.1±1.1# |
| $T_{chest}$ (˚C)^ | CON | 32.8±1.3 | 34.2±0.6 | 34.5±0.7 | 33.0±1.0 |
| | LBH | 32.4±0.8 | 24.5±3.0* | 21.3±3.2* | 28.7±2.0* |
| | WBH | 32.8±1.0 | 37.4±1.0# | 38.5±1.3# | 33.4±2.8# |

CON: control; LBH: lower-body heating with upper-body cooling; WBH: whole-body heating; $T_{rec:}$ rectal temperature; $T_{tympanic:}$ tympanic temperature; $T_{thigh:}$ thigh skin temperature; $T_{calf:}$ calf skin temperature; $T_{arm:}$ arm skin temperature; $T_{chest:}$ chest skin temperature; P: post.

^ time x condition interaction

* different from CON

\# different from other two conditions ($p<0.05$).

## Haematological measures

The haematological measures and adrenaline concentrations are shown in Table 3. An effect of time ($p<0.001$) and time x condition ($p<0.001$) but not condition ($p = 0.277$) was found for the platelet count. A higher platelet count was found immediately after WBH compared with the other conditions ($p<0.002$), while there was no difference between LBH and CON immediately post ($p = 0.436$). There was an effect of time ($p = 0.006$), but not of condition ($p = 0.084$) or time x condition ($p = 0.866$) for the total leukocyte count. Similarly, an effect of time was observed for monocytes ($p = 0.026$), but no effect of condition ($p = 0.645$) or a time x condition interaction ($p = 0.669$). For lymphocytes, there was an effect of time ($p = 0.001$) and a time x condition interaction ($p = 0.001$). At 60 min post-session, lymphocyte concentration was lower after WBH compared with the other two conditions ($p<0.001$). There was no effect of time ($p = 0.060$), condition ($p = 0.074$), nor a time x condition interaction effect ($p = 0.809$) for neutrophils. Plasma adrenaline concentration was higher following WBH compared with both other conditions ($p<0.001$) and following LBH compared with CON ($p = 0.027$).

## Brain-derived neurotrophic factor

The acute changes in serum and plasma BDNF concentration and BDNF release per platelet following the three experimental conditions are shown in Fig 1. There was no effect of time ($p = 0.116$) or condition ($p = 0.145$) for serum BDNF. However, there was a time x condition interaction effect ($p = 0.033$). Directly post-session, serum BDNF concentration was increased

**Table 2. Cardiovascular responses to the three experimental conditions.** Values are expressed as mean ± SD (N = 10).

| Parameter | Condition | | Time | | |
|---|---|---|---|---|---|
| | | **Pre** | **45 min** | **End** | **P+60 min** |
| HR (bpm)^ | CON | 58±15 | 59±13 | 55±11 | 56±13 |
| | LBH | 54±11 | 61±10 | 59±10 | 55±10 |
| | WBH | 59±15 | 90±16# | 110±15# | 62±17 |
| SBP (mmHg)^ | CON | 114±9 | 125±10 | 126±7 | 117±10 |
| | LBH | 120±7 | 130±13 | 133±11 | 134±6 |
| | WBH | 118±8 | 134±12# | 145±19# | 126±9 |
| DBP (mmHg)^ | CON | 63±5 | 65±8 | 68±5 | 67±5 |
| | LBH | 67±4 | 71±7 | 73±8 | 70±5 |
| | WBH | 64±6 | 64±4 | 70±7 | 63±6 |
| CO (L/min)^ | CON | 4.4±1.0 | - | 4.2±0.9 | - |
| | LBH | 4.1±0.8 | - | 4.4±1.1 | - |
| | WBH | 4.5±1.0 | - | 7.0±1.8# | - |
| Brachial BF (ml/min)^ | CON | 98±49 | - | 107±44 | - |
| | LBH | 111±83 | - | 60±28* | - |
| | WBH | 125±88 | - | 563±183# | - |
| Femoral BF (ml/min)^ | CON | 612±226 | 649±243 | 660±222 | - |
| | LBH | 609±226 | 842±261* | 943±349* | - |
| | WBH | 584±263 | 1427±332# | 1713±409# | - |
| Femoral SR (/s)^ | CON | 54.4±18.3 | 54.0±15.7 | 52.5±16.3 | - |
| | LBH | 49.2±14.2 | 65.1±21.5* | 68.6±27.4* | - |
| | WBH | 51.9±16.6 | 124.3±38.6# | 140.4±36.9# | - |
| Femoral VC (U)^ | CON | 6.8±2.1 | 7.0±1.8 | 6.9±1.9 | - |
| | LBH | 6.5±1.7 | 8.5±2.3* | 9.6±4.0* | - |
| | WBH | 6.2±2.1 | 15.5±2.4# | 17.4±3.6# | - |
| RPP (AU)^ | CON | 6819±2328 | 7241±1942 | 7368±1610 | 6747±1417 |
| | LBH | 6937±1645 | 7774±1322 | 7903±1060 | 7092±1263 |
| | WBH | 7194±2286 | 11946±2624# | 16403±3053# | 8172±2628# |
| PSI (AU)^ | CON | - | 0.04±0.18 | 0.02±0.48 | 0.04±0.38 |
| | LBH | - | 0.93±0.73* | 1.32±0.71* | 0.05±0.61 |
| | WBH | - | 2.79±1.06# | 5.66±1.17# | 1.71±0.73# |

CON: control; LBH: lower-body heating with upper-body cooling; WBH: whole-body heating; HR: heart rate; SBP: systolic blood pressure; DBP: diastolic blood pressure; CO: cardiac output; BF: blood flow; VC: vascular conductance; SR: shear rate; RPP: rate pressure product; PSI: physiological strain index; P: post.

^ time x condition interaction

* different from CON

# different from other two conditions ($p < 0.05$).

in WBH ($p = 0.044$), but not LBH ($p = 0.126$) or CON ($p = 0.454$). Serum BDNF concentration immediately after the session was higher in WBH and LBH compared with CON ($p = 0.048$), with no difference between WBH and LBH ($p = 0.206$). Plasma BDNF concentration showed an effect of time ($p < 0.001$) and condition ($p = 0.022$), as well as a time x condition interaction effect ($p = 0.001$). Immediately following the session, plasma BDNF concentration was increased in WBH ($p = 0.003$), but not LBH ($p = 0.468$) or CON ($p = 0.053$). There was a difference between WBH and CON ($p = 0.009$), but not between LBH and CON immediately after the session ($p = 0.134$). Finally, no effect of time ($p = 0.392$), condition ($p = 0.220$), or time x condition interaction ($p = 0.428$) was found for the BDNF content per platelet.

**Table 3. Adrenaline and haematological variables in responses to the three experimental conditions.** Values are expressed as mean ± SD (N = 10).

| Parameter | Condition | Pre | Post | P+60 min |
|---|---|---|---|---|
| Adrenaline (ng/mL)$^{\$\wedge}$ | CON | 12.2±7.9 | 16.0±10.0 | 19.3±14.0 |
| | LBH | 20.7±11.6 | 38.9±26.2* | 33.0±19.7 |
| | WBH | 15.6±7.5 | 70.9±30.5# | 13.8±7.3 |
| Leukocytes ($10^9$/L)$^{\$}$ | CON | 4.54±1.03 | 4.96±1.15 | 4.77±0.92 |
| | LBH | 4.79±0.77 | 5.29±0.85 | 5.11±0.81 |
| | WBH | 5.16±1.35 | 5.99±1.63 | 5.62±1.87 |
| Neutrophils ($10^9$/L) | CON | 2.41±0.67 | 2.65±0.87 | 2.65±0.75 |
| | LBH | 2.68±0.57 | 2.69±0.55 | 2.94±0.62 |
| | WBH | 3.09±1.35 | 3.35±1.35 | 3.67±1.51 |
| Monocytes ($10^9$/L)$^{\$}$ | CON | 0.40±0.14 | 0.42±0.13 | 0.39±0.11 |
| | LBH | 0.40±0.13 | 0.44±0.13 | 0.39±0.13 |
| | WBH | 0.40±0.12 | 0.46±0.15 | 0.42±0.17 |
| Lymphocytes ($10^9$/L)$^{\$\wedge}$ | CON | 1.49±0.33 | 1.61±0.34 | 1.50±0.25 |
| | LBH | 1.44±0.25 | 1.67±0.37 | 1.54±0.25 |
| | WBH | 1.46±0.38 | 1.37±0.75# | 1.27±0.29# |
| Platelets ($10^9$/L)$^{\$\wedge}$ | CON | 199±20 | 205±25 | 207±22 |
| | LBH | 209±33 | 211±31 | 209±38 |
| | WBH | 189±27 | 235±19# | 197±36 |
| Δ Plasma volume (-fold)^ | CON | N/A | 1.06±0.11# | 1.05±0.06# |
| | LBH | N/A | 0.94±0.07 | 0.95±0.06 |
| | WBH | N/A | 0.92±0.06 | 0.95±0.08 |

Abbreviations: CON: control; LBH: lower-body heating with upper-body cooling; WBH: whole-body heating.

$ effect of time

^ time x condition interaction

* different from CON

# different from other two conditions ($p<0.05$).

## Correlations

There was a strong, positive correlation between the change in serum BDNF concentration and BDNF per platelet ($r = 0.90$, $p<0.001$). There was no correlation between the change in serum BDNF concentration and plasma BDNF concentration ($r = 0.43$, $p = 0.24$), platelet count ($r = 0.07$, $p = 0.861$) or the physiological measures ($\Delta T_{rec}$ $r = 0.06$, $p = 0.846$; $\Delta$femoral artery shear rate $r = -0.16$, $p = 0.652$, $\Delta$HR $r = -0.15$, $p = 0.689$). The acute change in plasma BDNF concentration was not correlated with the change in platelet BDNF ($r = 0.05$, $p = 0.900$), platelet count ($r = 0.47$, $p = 0.203$) or any of the physiological measures ($\Delta T_{rec}$ $r = 0.14$, $p = 0.720$; $\Delta$femoral artery shear rate $r = 0.07$, $p = 0.861$, $\Delta$HR $r = 0.17$, $p = 0.665$). Platelet BDNF was not correlated with platelet count ($r = -0.31$, $p = 0.379$) or any of the physiological measures ($\Delta T_{rec}$ $r = -0.07$, $p = 0.841$; $\Delta$femoral artery shear rate $r = 0.02$, $p = 0.941$, $\Delta$HR $r = -0.36$, $p = 0.309$).

## Discussion

This study investigated the efficacy of WBH as well as a local heating protocol to induce an acute BDNF response. Upper-body cooling during passive heating reduced cardiac output, the rate pressure product as well as the physiological strain index when compared with WBH. However, while WBH acutely elevated plasma and serum BDNF concentration, this response

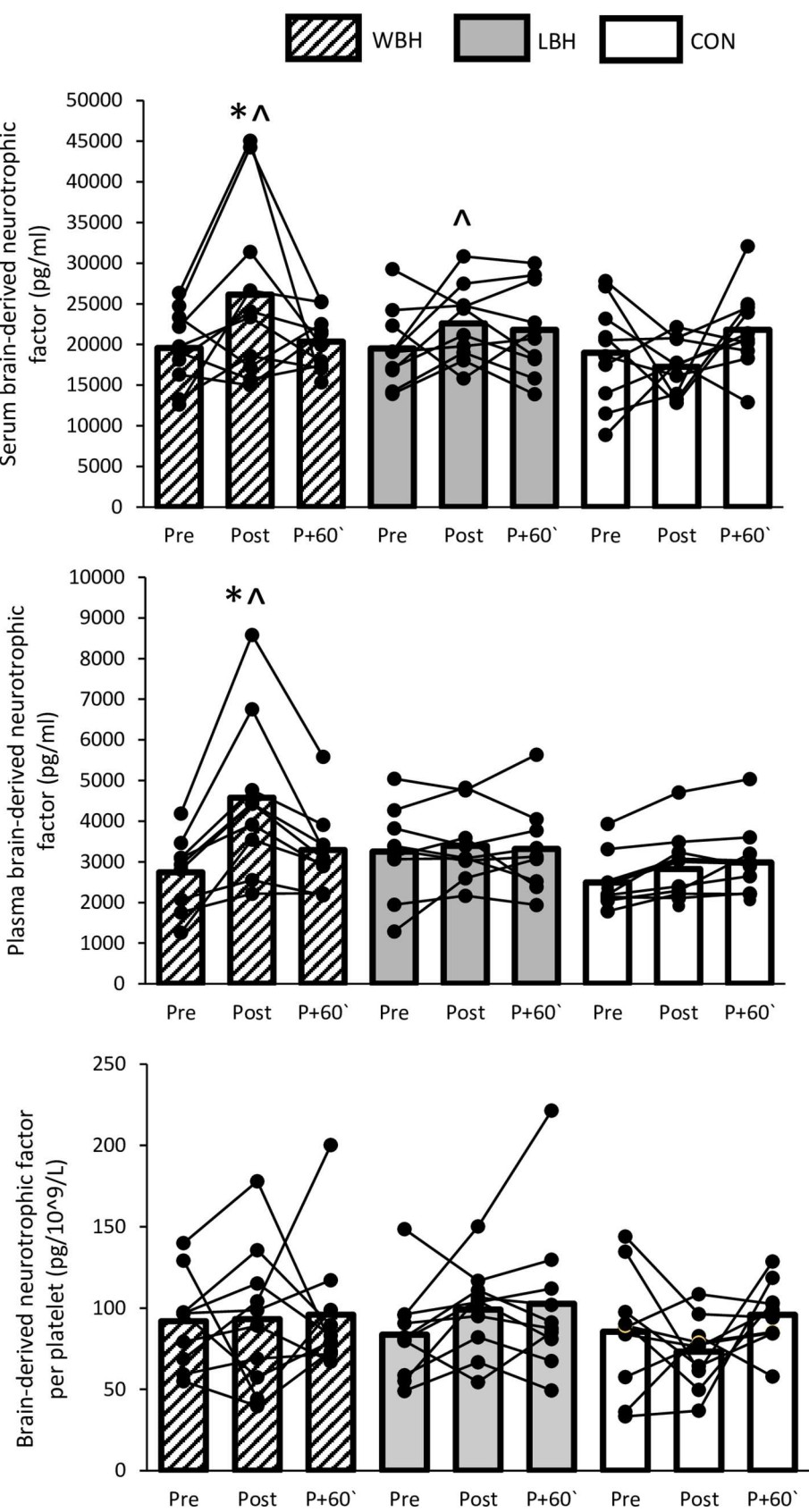

**Fig 1. Brain-derived neurotrophic factor responses in serum and plasma following the three experimental conditions.** * Different from Pre, ^ different from CON ($p<0.05$).

was blunted after LBH, despite an increase in sympathetic activity and femoral artery shear rate.

Passively elevating body temperature by WBH induced an acute increase in circulating BDNF, corroborating the findings from Kojima et al. [11], who reported an increase in serum BDNF concentration following 20 min of hot water immersion. The effect of an elevated body temperature on peripheral BDNF may be mediated by a range of factors acting on multiple cellular sources of BDNF. For instance, exercising in a warm environment has been shown to increase S100β, a marker for blood-brain-barrier permeability [34], which may increase the contribution of the brain to plasma BDNF [17]. Conversely, Kojima et al. [11] found significant increases in serum BDNF concentration without concomitant changes in S100β following 20 min of hot water immersion. In support, Shepley et al. [35] recently reported that 60 min of moderate-to-severe hyperthermia induced by hot water immersion (+2˚C core temperature) has a negligible impact on biomarkers of neurovascular integrity and permeability released from the brain. This highlights the fact that sources other than the brain produce and release BDNF in response to various physiological stimuli [17].

As an example of such a physiological stimulus, the vascular endothelium produces and secretes BDNF in response to shear stress *in vitro* [19], and the expression of the BDNF-receptor TrkB on endothelial cells has led others to suggest that a positive feedback loop exists in which the binding of circulating BDNF with TrkB activates BDNF production by endothelial cells [17]. WBH induced a ~3- and ~4-fold increase in femoral artery and brachial artery blood flow, respectively, potentially contributing to BDNF release by endothelial cells in the vasculature. Further, the increase in cardiac output and adrenaline indicates an increase in sympathetic activity during WBH. Walsh et al. [20] support the importance of sympathetic activity for the increase in circulating BDNF concentration. By investigating handgrip exercise, the authors exploited the notion that local metabolic stress within skeletal muscle during exercise appears more important than the absolute muscle mass involved for sympathetic activation [36]. Targeting sympathetic activity by this small muscle mass exercise increased serum BDNF concentration, despite the limited metabolic cost and body temperature rise associated with the activity. The effect of sympathetic activity on serum BDNF concentration may have been mediated by the recruitment of platelets from the spleen into the circulation that occurred during WBH, indicated by the increase in platelet count. However, contrary to this notion, there was no correlation between the change in platelet count and serum BDNF concentration following WBH. Indeed, the relationship between circulating platelets and BDNF is impacted by more than platelet count *per se*, as aspects of platelet function are altered by passive heating in humans [37] or in pathological states like major depressive disorder [38]. In addition, as for the lack of significant correlations between the BDNF response and measures of blood flow and sympathetic activity, it should be noted that this study included a relatively small sample size and was not designed to explore correlations between these outcome measures.

In contrast to the acute increase in serum and plasma BDNF concentration after WBH, limiting the rise in $T_{rec}$ by the localised cooling used in LBH blunted this response. In the exercise literature, there appears to be a dose-response relationship between exercise intensity and the acute increase in BDNF concentration [7, 39]. For instance, a systematic review showed that 69% of studies investigating a high-intensity exercise protocol reported an acute increase in BDNF concentration, while this was only 44% in studies on low and moderate-intensity

exercise [39]. This could be explained by the larger increase in shear stress [40] and sympathetic activity [41] during high compared with moderate-intensity exercise. As the recently put forward exercise intensity threshold for the elevation of BDNF concentration [42] may thus be related to these factors, it is likely that the increase in shear stress and sympathetic activity by LBH was not sufficient to elevate BDNF concentration. Indeed, although LBH induced a small increase in adrenaline concentration, none of the other measures of sympathetic activity were elevated (e.g. cardiac output, diastolic blood pressure); despite the large increase in $T_{skin}$ of the lower limbs. This underscores the relatively large contribution of $T_{core}$ when compared with $T_{skin}$ to sympathetic activity-related processes such as changes in vasomotor activity and catecholamine production [43]. In line with the limited increase in sympathetic activity, LBH resulted in a 40% rise in shear rate compared with nearly 300% in WBH. Future attempts to create a tolerable passive heating protocols to elevate circulating BDNF concentration may thus need to induce a larger increase in $T_{core}$; of which the exact magnitude will depend on the balance between the attendant cardiovascular strain and the acute BDNF response. As such, heating the lower limbs in the absence of upper-body cooling may be an appealing protocol to test in future studies.

In the present study, BDNF concentration after WBH was elevated in serum as well as plasma. This suggests that passively elevating body temperature stimulates BDNF release by peripheral tissues (primarily reflected by plasma measurements) as well as platelets (primarily reflected by serum measurements). Interestingly, in contrast to Kojima et al. [11], the increase in serum BDNF concentration found after WBH in the present study was accompanied by an elevated platelet count. Although the lack of BDNF mRNA expression in platelets suggest that these cells do not synthesise BDNF *de novo* [44], BDNF stored in platelets is released during the clotting process of the serum collection procedure [45]. *In vitro* experiments suggest that shear stress and sympathetic activation can enhance BDNF release per platelet [18]. However, there was no change in BDNF release per platelet following WBH, suggesting that the elevated serum BDNF concentration in the present study was mainly the result of an increase in platelet count. On the other hand, despite no change in BDNF release per platelet in WBH, there was a strong correlation ($r = 0.90$) between the change in serum BDNF concentration and BDNF per platelet. Regardless, it should be noted that the calculated BDNF per platelet is derived from an indirect method with inbuilt assumptions about the sources of BDNF in plasma and serum [32]. Future studies could employ direct biochemistry techniques to assess BDNF content and release by platelets in response to physiological stress to further investigate the role of platelets in the concentration of circulating BDNF [37, 38].

## Practical applications and future directions

The acute elevation of BDNF concentration following WBH provides strong rationale to further investigate the efficacy of passive heating protocols to elevate BDNF concentration and improve cognitive function. At the same time, whilst more research on the safety of heat therapy is warranted, it should be noted that whole-body passive heat stress may be contraindicated for some individuals. For example, an observational report noted the occurrence of a cardiac arrest during hot water bathing in 9.84 out of 100,000 people, while the level of consciousness after an adverse event in the bath was negatively related to the core temperature attained [46]. In addition, whole-body heating is associated with higher thermal discomfort and more negative affective responses compared with local heating [21]. As such, to confidently prescribe effective and safe passive heating protocols for a wide range of (clinical) populations (e.g., older adults and persons with chronic heart failure), further research into passive heating protocols with a reduced cardiovascular strain and thermal discomfort is needed.

Aside from exploring additional tolerable and yet effective protocols, chronic intervention studies could build on Glazachev et al. [12] to further investigate the effects of repeated passive heating on BDNF concentration and cognitive function. Importantly, such studies should focus on persons at risk for reduced BDNF expression or cognitive function due to metabolic dysfunction or impaired mobility. While such individuals may arguably benefit most from interventions that elevate BDNF concentration, physiological responses to heat stress can be impacted by old age [15] and health conditions such as type II diabetes mellitus [16]; reinforcing the need for studies in specific populations.

In conclusion, while the local cooling applied in LBH reduces cardiovascular strain when compared with WBH, this protocol does not elevate circulating BDNF concentration. A larger increase in shear rate and sympathetic activity, potentially through elevating $T_{core}$ to a larger extent than in LBH, may thus be needed to induce an acute BDNF response through passive heating. In contrast, the acute increase in plasma and serum BDNF concentration following WBH provides further support to pursue research into the potential of passive heat therapy to elevate circulating BDNF concentration.

## Acknowledgments

The authors thank Prof. George Havenith and Dr Alex Lloyd, who kindly provided the water-perfused suit. Further, the authors thank Miguel Dos Santos, Greg Handsley and Christian Andersen for their excellent assistance during data collection. The authors acknowledge the support of the National Institute for Health Research (NIHR) Leicester Biomedical Research Centre, as well as the Kyoten Research Center of Sports for Persons with Impairments. The views expressed are those of the authors and not necessarily those of the NHS, the NIHR, the Department of Health, nor those of the Kyoten Research Center of Sports for Persons with Impairments.

## Author Contributions

**Conceptualization:** Takahiro Ogawa, Sven P. Hoekstra, Yoshi-Ichiro Kamijo, Victoria L. Goosey-Tolfrey, Christof A. Leicht.

**Data curation:** Takahiro Ogawa, Sven P. Hoekstra.

**Formal analysis:** Takahiro Ogawa, Sven P. Hoekstra, Jeremy J. Walsh, Christof A. Leicht.

**Funding acquisition:** Yoshi-Ichiro Kamijo, Victoria L. Goosey-Tolfrey, Fumihiro Tajima F., Christof A. Leicht.

**Investigation:** Takahiro Ogawa, Sven P. Hoekstra, Christof A. Leicht.

**Methodology:** Takahiro Ogawa, Sven P. Hoekstra, Yoshi-Ichiro Kamijo, Victoria L. Goosey-Tolfrey, Jeremy J. Walsh, Christof A. Leicht.

**Project administration:** Takahiro Ogawa, Sven P. Hoekstra, Victoria L. Goosey-Tolfrey, Christof A. Leicht.

**Resources:** Yoshi-Ichiro Kamijo, Victoria L. Goosey-Tolfrey, Fumihiro Tajima F., Christof A. Leicht.

**Software:** Takahiro Ogawa, Sven P. Hoekstra, Jeremy J. Walsh.

**Supervision:** Yoshi-Ichiro Kamijo, Victoria L. Goosey-Tolfrey, Fumihiro Tajima F., Christof A. Leicht.

**Validation:** Jeremy J. Walsh, Fumihiro Tajima F., Christof A. Leicht.

**Visualization:** Takahiro Ogawa, Sven P. Hoekstra, Jeremy J. Walsh, Fumihiro Tajima F.

**Writing – original draft:** Takahiro Ogawa, Sven P. Hoekstra.

**Writing – review & editing:** Takahiro Ogawa, Sven P. Hoekstra, Yoshi-Ichiro Kamijo, Victoria L. Goosey-Tolfrey, Jeremy J. Walsh, Fumihiro Tajima F., Christof A. Leicht.

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
