## [Decision Letter · Decision Letter 0]

18 Jun 2021

PONE-D-21-13550

Serum and plasma brain-derived neurotrophic factor concentration are elevated by systemic but not local passive heating

PLOS ONE

Dear Dr. Hoekstra,

Thank you for submitting your manuscript to PLOS ONE. After careful consideration, we feel that it has merit but does not fully meet PLOS ONE’s publication criteria as it currently stands. Therefore, we invite you to submit a revised version of the manuscript that addresses the points raised during the review process.

Please respond to the reviewers comments with detailed responses.

We look forward to receiving your revised manuscript.

Kind regards,

Caroline Sunderland

Academic Editor

PLOS ONE

Journal Requirements:

2. Thank you for submitting the above manuscript to PLOS ONE. During our internal evaluation of the manuscript, we found significant text overlap between your submission and the following previously published works, some of which you are an author.

https://cdnsciencepub.com/doi/10.1139/apnm-2020-0704

https://cdnsciencepub.com/doi/10.1139/apnm-2020-0704

Please revise the manuscript to rephrase the duplicated text, cite your sources, and provide details as to how the current manuscript advances on previous work. Please note that further consideration is dependent on the submission of a manuscript that addresses these concerns about the overlap in text with published work.

Additional Editor Comments:

Reviewers' comments:

Reviewer's Responses to Questions

**Comments to the Author**

1. Is the manuscript technically sound, and do the data support the conclusions?

Reviewer #1: Yes

Reviewer #2: Yes

2. Has the statistical analysis been performed appropriately and rigorously? 

Reviewer #1: Yes

Reviewer #2: Yes

3. Have the authors made all data underlying the findings in their manuscript fully available?

Reviewer #1: Yes

Reviewer #2: Yes

4. Is the manuscript presented in an intelligible fashion and written in standard English?

Reviewer #1: Yes

Reviewer #2: Yes

5. Review Comments to the Author

Reviewer #1: This study evaluates the response of circulatory BDNF to different heating strategies, all using a water perfused suit but targeting different areas of the body. Findings suggest the impact of whole body heating on circulating BDNF concentration is greater in comparison to lower body heating alone, highlighting the need for greater internal heat alterations to influence BDNF. Rectal temperature was greater in the lower body heating protocol when compared to the control, however this was not great enough to influence BDNF.

This study generates a strong rational for the need to know the impact of passive heat stress, based on the assumption this would be a better option for improving health outcomes for those populations unable exercise, a well supported method of improving BDNF responses. This study adds to the current literature where local heating strategies have not previously been assessed in relation to BDNF.

Although this study highlights a strong rational around the benefits for use in populations who can’t exercise, it does not detail any of the potential side effects/negative implications. Having an understanding of how the target population’s discomfort and affect is influenced may provide greater application and understanding when the research proceeds to use it in the desired population.

There are no major issues with this study and it’s design. However, some minor recommendations detailed below:

Specifics:

An excellent overview of BDNF and it’s various effects are detailed. An extensive mechanistic approach with complex measurements are included in the study.

Providing a more in depth evaluation of the populations this strategy applies to would strengthen your rational.

Dinooff et al (2016) – what types of exercise? In humans? Elaborate.

General – add specifics around types of exercise and heat stress used for literature within the introduction.

Good rational provided based on why other protocols are not applicable.

Methods would benefit from more detail on water perfused suit, specifically how the upper body cooling/lower body heating protocol was implemented. Including a figure (supplementary material) would be beneficial.

Upper body cooling is used in the local heating strategy – what implications does this have compared to if a thermoneutral option was used there. Worth discussing why thermoneutral was not used in upper body with lower body heating. (This may be better understood when methods of the suit are elaborated on).

Were individual differences in sweat rate accounted for? What is the rational for the drink volume used?

Were any discomfort measures or measures of affect taken? Practical application – is it feasible to use in elderly, disabled or individuals with disease without knowledge of the levels of discomfort and the influence on feeling/affect in a healthy population.

Shear stress is discussed throughout when referring to the potential mechanisms involved – this discussion could be strengthened through a more in depth introduction to this earlier in the paper.

All tables and figures are relevant and well formatted.

Reviewer #2: The manuscript by Ogawa et. al. titled, “Serum and plasma brain-derived neurotrophic factor concentration are elevated by systemic by not local passive heating” determines the impact of whole body vs lower body heating on BDNF and cardiovascular parameters. This study appears to be carefully conducted and prepared. I offer the following comments for the authors to consider.

Abstract:

• I know space is limed, but the abstract would be enhanced by including some of the actual data instead of just p-values. Perhaps at least for some of you main dependent variables.

Introduction:

• In the second paragraph, second sentence: I believe you mean is and not in (…especially in situations where exercise is not accessible….

• I believe there is some literature that indicates no difference in the exercise response between different environmental temperatures (Collins et al, 2017, Temperature). I just wonder if this would be useful here or in the discussion to help set up the notion that the BDNF response is likely more related to cardiovascular strain than the temperature??? Don’t feel obligated on this comment as I think it is adequately presented as is, but it may be worth considering??

Methods:

• For the food diary, was the diet of the first trial replicated in the other two. Please provide some info on how this was used.

• Why was the CON at 36°C? Throughout the manuscript, I don’t really like the word thermoneutral. Is a 36°C skin temperature what you consider thermoneutral. I was finding thermoneutral skin temp to be 33-35 and other things indicating 28-32 air temperature? Perhaps some justification/references are needed here. I fear that thermoneutral may mean different things to different people and the working should be changed or justified. I don’t have a problem with the use of 36 C, just perhaps tweaking the presentation.

• You report the temp and humidity of the lab. I commend you for adding this detail. Perhaps add a p-value from statistical analysis so that the reader can more readily and quickly interpret this info.

• I was having a hard time finding your temperature sensors based on the given information. Please provide a product # or name? Is the zero-heat flux temp senor validated for this use and is there a reference to include? Was there a specific data logger that you used? I was just trying to picture the exact set-up and was having a hard time. I think that by adding more detail here it would enhance this section.

• How did you determine the water volume provided? It doesn’t appear to be based on the actual weight change of the subject?

• Statistics: I would suggest using “Fishers Protected LSD method” verbiage for describing your post-hoc. It is exactly what you did with the t-test, but the verbiage helps remind the reader that your error rate is accounted within the ANOVA itself. Not a big deal, just something that I find helps if others (reviewers or readers) criticize this approach.

Discussion:

• My interpretation of the discussion is that the BDNF response is not directly related to the heat, but rather indirectly via the cardiovascular stress which is supported by the data. This is a bit difficult based on the lack of correlation between BDNF and cardiovascular stress parameters. Could skin thermal receptors play a role here by activating the sympathetic nervous system. The thought here would be that the WBH activates more of the receptors than the LBH and lead to the differences? This comment is probably beyond the scope of your project, but may be something to consider as you think about the potential mechanism in light of the lack of correlations with some of the cardiovascular correlations. I think it is important to further discuss these lack of correlations. I just kept thinking of technical issues around the use of correlations (relatively low sample size, having 3 temperature conditions instead a larger range, ect.) and what other factors may stimulate this alteration. I wouldn’t do too much here, but you may want to address the correlations a bit more.

6. PLOS authors have the option to publish the peer review history of their article (what does this mean?). If published, this will include your full peer review and any attached files.

Reviewer #1: No

Reviewer #2: No

---

## [Author Response · Author response to Decision Letter 0]

14 Jul 2021

Dear Reviewer 1,

Thank you for the careful reading of our manuscript and the insightful comments. We also appreciate your comment regarding the potentially negative impact of (whole-body) passive heating for certain populations. We have tried to incorporate your suggestions and believe they have improved the quality of the manuscript.

This study evaluates the response of circulatory BDNF to different heating strategies, all using a water perfused suit but targeting different areas of the body. Findings suggest the impact of whole body heating on circulating BDNF concentration is greater in comparison to lower body heating alone, highlighting the need for greater internal heat alterations to influence BDNF. Rectal temperature was greater in the lower body heating protocol when compared to the control, however this was not great enough to influence BDNF. This study generates a strong rational for the need to know the impact of passive heat stress, based on the assumption this would be a better option for improving health outcomes for those populations unable exercise, a well supported method of improving BDNF responses. This study adds to the current literature where local heating strategies have not previously been assessed in relation to BDNF.

Thank you for the kind words.

Although this study highlights a strong rational around the benefits for use in populations who can’t exercise, it does not detail any of the potential side effects/negative implications. Having an understanding of how the target population’s discomfort and affect is influenced may provide greater application and understanding when the research proceeds to use it in the desired population.

We agree that this is an important consideration. This aligns closely to other comments by you and the other reviewer. As such, we have incorporated discussion on the implications for specific populations and the potential thermal discomfort during whole-body heating in a “practical application and future directions” paragraph at the end of the Discussion. 

There are no major issues with this study and it’s design. However, some minor recommendations detailed below:

Specifics:

An excellent overview of BDNF and it’s various effects are detailed. An extensive mechanistic approach with complex measurements are included in the study.

Thank you for the kind words on the Introduction.

Providing a more in depth evaluation of the populations this strategy applies to would strengthen your rational.

Thank you for this comment. 

We agree that this is an important consideration. We have now added a sentence on the potential implications of an effective, but more tolerable passive heating protocol in the Introduction (Line 95). As highlighted in the Introduction, this can make this intervention more accessible to people with chronic heart failure and older adults for example. Additionally, as we have previously shown that a local heating protocol can lead to more favourable perceptual responses (PMID: 33439769), we believe the implications of such a protocol may extend to all individuals using heat therapy for health promotion. As mentioned above, these considerations are now also included in the Discussion (Line 363). 

Dinooff et al (2016) – what types of exercise? In humans? Elaborate.

We have now made it clear that this is primarily aerobic exercise, and that Dinoff et al (2016) has provided evidence for this effect in humans. We have kept it brief, as passive heat stress was the primary focus of the manuscript (see also the comment below).

General – add specifics around types of exercise and heat stress used for literature within the introduction.

We have now added specifics related to the heat stress described, as this is the focus of the manuscript (e.g. Line 74, Line 292).

Good rational provided based on why other protocols are not applicable.

Thank you.

Methods would benefit from more detail on water perfused suit, specifically how the upper body cooling/lower body heating protocol was implemented. Including a figure (supplementary material) would be beneficial. 

Thank you for this comment and we agree that this suggestion would clarify the intervention for the reader. We have published on other outcome measures in response to the protocol employed in the current study and have now more clearly referred the reader to that publication (Line 122; PMID: 33439769), as a detailed outline has been provided in that article.

Upper body cooling is used in the local heating strategy – what implications does this have compared to if a thermoneutral option was used there. Worth discussing why thermoneutral was not used in upper body with lower body heating. (This may be better understood when methods of the suit are elaborated on).

The reason for the upper-body cooling instead of a thermoneutral upper body condition was that we wanted to investigate a condition in which core temperature rises were limited, whilst a local heat stress was applied (a more substantial rise in core temperature was likely to occur with heating of the legs in combination with a thermoneutral condition for the upper body (PMID: 30303416)). Nonetheless, considering the absence of an acute BDNF response after the LBH protocol in the current study, the suggested combination of heating the legs without cooling the upper body would be an interesting protocol to investigate in the future.

Were individual differences in sweat rate accounted for? What is the rational for the drink volume used?

The water volume provided during the sessions was based on pilot work. We have not taken into account individual differences in sweat rate, and agree that this would have been a good addition to the study. However, considering the lack of difference in body mass loss and plasma volume change between the conditions, we believe that the potential effect of hydration status on BDNF concentration (PMID: 28828079) was likely to have been negligible in the current study.

Were any discomfort measures or measures of affect taken? Practical application – is it feasible to use in elderly, disabled or individuals with disease without knowledge of the levels of discomfort and the influence on feeling/affect in a healthy population.

Thank you for this relevant comment. As mentioned previously, we have published on other outcome measures in response to the same experimental protocols. The perceptual responses to the heating protocols were one of the main outcome measures of that article (PMID: 33439769), including measures of affect, thermal comfort and thermal sensation. We have included discussion and a reference to this article in the newly added “practical implications and future directions” paragraph of the Discussion.

Shear stress is discussed throughout when referring to the potential mechanisms involved – this discussion could be strengthened through a more in depth introduction to this earlier in the paper.

We have now included discussion on shear stress as a potential stimulator of BDNF production and how it provides rationale to investigate local passive heating protocols in the Introduction section (Line 95).

All tables and figures are relevant and well formatted.

Thank you.

Dear Reviewer 2,

Thank you for your insightful comments and suggestions to improve the quality of the manuscript. We are also grateful for your valuable suggestions as to what the mechanisms underpinning the BDNF response to heat stress may be. We believe that the changes we have made based on your comments have improved the quality of the manuscript.

The manuscript by Ogawa et. al. titled, “Serum and plasma brain-derived neurotrophic factor concentration are elevated by systemic by not local passive heating” determines the impact of whole body vs lower body heating on BDNF and cardiovascular parameters. This study appears to be carefully conducted and prepared. I offer the following comments for the authors to consider.

Abstract:

• I know space is limed, but the abstract would be enhanced by including some of the actual data instead of just p-values. Perhaps at least for some of you main dependent variables.

Thank you for this comment and we agree that this strengthens the abstract. We have now included the serum and plasma BDNF data.

Introduction:

• In the second paragraph, second sentence: I believe you mean is and not in (…especially in situations where exercise is not accessible….

We have now changed in to is.

• I believe there is some literature that indicates no difference in the exercise response between different environmental temperatures (Collins et al, 2017, Temperature). I just wonder if this would be useful here or in the discussion to help set up the notion that the BDNF response is likely more related to cardiovascular strain than the temperature??? Don’t feel obligated on this comment as I think it is adequately presented as is, but it may be worth considering??

This is an interesting suggestion. Indeed, one could argue that as long as for instance shear stress is elevated (through exercise, for example), an additional increase in temperature may not be necessary to increase BDNF concentration. However, the lack of such cardiovascular measurements in the exercise studies (Collings et al. 2017; Goekint et al., 2011) makes it difficult to uncouple the effect of hyperthermia and e.g. shear stress with any confidence. Moreover, there is evidence for the contrary (i.e. larger BDNF response in hot environment (Goekint et al., 2011; PMID: 21385602)), and closer inspection of the data presented in Collins et al. (2017) shows that BDNF concentration is ~25% higher after exercise in the heat compared with the cold and moderate conditions; suggesting that the lack of effect in that study may have been related to limited statistical power. Therefore, also for the sake of clarity in the Introduction, we prefer to not include Collins et al. (2017) in this section. However, we agree with your interpretation that the BDNF response is likely closer related to factors such as shear stress and sympathetic activation than hyperthermia per se, and have tried to emphasise that further based on your comments (e.g. Line 95 - 98).

Methods:

• For the food diary, was the diet of the first trial replicated in the other two. Please provide some info on how this was used.

Thank you for this comment. We have now described this process in more detail (Line 121).

• Why was the CON at 36°C? Throughout the manuscript, I don’t really like the word thermoneutral. Is a 36°C skin temperature what you consider thermoneutral. I was finding thermoneutral skin temp to be 33-35 and other things indicating 28-32 air temperature? Perhaps some justification/references are needed here. I fear that thermoneutral may mean different things to different people and the working should be changed or justified. I don’t have a problem with the use of 36 C, just perhaps tweaking the presentation.

Thank you for this comment. We used this condition as water-perfused suit temperatures in the range of 33-36°C can keep core temperature stable (PMID: 30303416; PMID: 16763078). Pilot work in preparation for this study showed that water-perfused temperatures below 36°C resulted in a core temperature reduction. As such, this temperature was chosen as the control condition.

Despite this rationale, we agree that this is not truly thermoneutral, as the suit temperature was substantially higher than resting skin temperature. Therefore, we have removed the term thermoneutral and have changed this to “control condition” throughout the manuscript.

• You report the temp and humidity of the lab. I commend you for adding this detail. Perhaps add a p-value from statistical analysis so that the reader can more readily and quickly interpret this info.

Thank you for this comment. We have now included the smallest p-value of the one-way ANOVAs conducted on the comparisons for temperature and humidity (p>0.452).

• I was having a hard time finding your temperature sensors based on the given information. Please provide a product # or name? Is the zero-heat flux temp senor validated for this use and is there a reference to include? Was there a specific data logger that you used? I was just trying to picture the exact set-up and was having a hard time. I think that by adding more detail here it would enhance this section.

Thank you for this comment. The Bair Hugger sensor is a coin-sized sensor that was placed on the vastus lateralis and covered with a small Tupperware box. The sensor has been validated previously, and we have now included a reference for this method (Binzoni et al., 1999). We hope that this reference, together with the reference to the product and manufacturer already provided, will give the reader sufficient information to visualise and evaluate this outcome measure.

More generally, we have now clearer referred the reader to a manuscript we have previously published, describing a different set of outcome measures with the same experimental procedures (Hoekstra et al., 2021; APNM). Here the experimental set-up is described in more detail, and the readers can refer to this article if more specific information is required. This article also includes a detailed image of the experimental set-up.

• How did you determine the water volume provided? It doesn’t appear to be based on the actual weight change of the subject?

Thank you for this comment. The water provided was based on the mean body mass loss that we observed in several participants (N=4) during our pilot testing. We agree that we did not assess individual sweat rate for each participant separately, and thus we did not achieve exact euhydration in each trial. However, considering the lack of statistical difference in body mass loss between trials, we believe we have managed to avoid the potentially confounding effect of dehydration in the present study. 

• Statistics: I would suggest using “Fishers Protected LSD method” verbiage for describing your post-hoc. It is exactly what you did with the t-test, but the verbiage helps remind the reader that your error rate is accounted within the ANOVA itself. Not a big deal, just something that I find helps if others (reviewers or readers) criticize this approach.

Thank you for this suggestion. We have now changed this in the statistical analysis section.

Discussion:

• My interpretation of the discussion is that the BDNF response is not directly related to the heat, but rather indirectly via the cardiovascular stress which is supported by the data. This is a bit difficult based on the lack of correlation between BDNF and cardiovascular stress parameters. Could skin thermal receptors play a role here by activating the sympathetic nervous system. The thought here would be that the WBH activates more of the receptors than the LBH and lead to the differences? This comment is probably beyond the scope of your project, but may be something to consider as you think about the potential mechanism in light of the lack of correlations with some of the cardiovascular correlations. I think it is important to further discuss these lack of correlations. I just kept thinking of technical issues around the use of correlations (relatively low sample size, having 3 temperature conditions instead a larger range, ect.) and what other factors may stimulate this alteration. I wouldn’t do too much here, but you may want to address the correlations a bit more.

Thank you for your accurate interpretation of our discussion of the data; we agree that it is indeed likely that the BDNF response is more closely linked to some of the physiological consequences of heat stress than hyperthermia itself (see e.g. PMID: 12008958). This is something we have tried to lay out in the paragraph starting at Line 297. However, we agree that the lack of correlations between the BDNF response and such physiological markers is somewhat surprising. In line with your note of caution, we have now included a sentence on the limitations of the correlation analyses in the present study (Line 316).

---

## [Decision Letter · Decision Letter 1]

17 Nov 2021

Serum and plasma brain-derived neurotrophic factor concentration are elevated by systemic but not local passive heating

PONE-D-21-13550R1

Dear Dr. Hoekstra,

We’re pleased to inform you that your manuscript has been judged scientifically suitable for publication and will be formally accepted for publication once it meets all outstanding technical requirements.

Kind regards,

Caroline Sunderland

Academic Editor

PLOS ONE

Additional Editor Comments (optional):

Reviewers' comments:

Reviewer's Responses to Questions

**Comments to the Author**

1. If the authors have adequately addressed your comments raised in a previous round of review and you feel that this manuscript is now acceptable for publication, you may indicate that here to bypass the “Comments to the Author” section, enter your conflict of interest statement in the “Confidential to Editor” section, and submit your "Accept" recommendation.

Reviewer #1: All comments have been addressed

2. Is the manuscript technically sound, and do the data support the conclusions?

Reviewer #1: Yes

3. Has the statistical analysis been performed appropriately and rigorously? 

Reviewer #1: Yes

4. Have the authors made all data underlying the findings in their manuscript fully available?

Reviewer #1: Yes

5. Is the manuscript presented in an intelligible fashion and written in standard English?

Reviewer #1: Yes

6. Review Comments to the Author

Reviewer #1: The authors have made adequate amendments to the paper to warrant publication. Greater clarity has bee provided in specific areas and all comments have been addressed.

7. PLOS authors have the option to publish the peer review history of their article (what does this mean?). If published, this will include your full peer review and any attached files.

Reviewer #1: No

---

## [Editor Report · Acceptance letter]

1 Dec 2021

PONE-D-21-13550R1 

Serum and plasma brain-derived neurotrophic factor concentration are elevated by systemic but not local passive heating 

Dear Dr. Hoekstra:

I'm pleased to inform you that your manuscript has been deemed suitable for publication in PLOS ONE. Congratulations! Your manuscript is now with our production department. 

Kind regards, 

on behalf of

Dr. Caroline Sunderland 

Academic Editor

PLOS ONE